# A Review of Anthropometric Measurements for Saudi Adults and Elderly, Directions for Future Work and Recommendations to Establish Saudi Guidelines in Line with the Saudi 2030 Vision

**DOI:** 10.3390/healthcare11141982

**Published:** 2023-07-08

**Authors:** Essra A. Noorwali, Abeer M. Aljaadi

**Affiliations:** Department of Clinical Nutrition, Faculty of Applied Medical Sciences, Umm Al-Qura University, Makkah 21955, Saudi Arabia

**Keywords:** anthropometry, Saudi Arabia, adults, noncommunicable diseases, obesity

## Abstract

Body weight is a significant risk factor for the disease burden of noncommunicable diseases (NCDs). Anthropometric measurements are the first step in determining NCDs risk, and clinicians must have access to valid cutoffs. This study aims to review the literature of Saudi national guidelines and studies previously conducted in Saudi Arabia (SA) and to provide insights and recommendations to establish national guidelines in anthropometric measurements for Saudi adults/elderly in line with the Saudi 2030 Vision. In total, 163 studies were included, and 12 of them contributed to the development of specific anthropometric cutoffs. Cutoffs for metabolic syndrome, waist circumference, and body mass index were established in Saudi adults. However, limited studies were conducted in the elderly. This review warrants establishing standard cutoffs of Saudi adult anthropometrics to avoid over/underreporting of malnutrition and adiposity. This review will help policymakers and the Ministry of Health to establish national guidelines and standard cutoffs to be used in SA for anthropometric measurements that may assist in detecting malnutrition and NCDs.

## 1. Introduction

Noncommunicable diseases (NCDs) are chronic diseases, such as diabetes, cardiovascular diseases (CVDs), chronic respiratory diseases, and cancers. NCDs are caused by a combination of genetic, environmental, physiological, and behavioral factors and kill 41 million people each year, equivalent to 71% of all deaths globally [1]. In addition, NCDs pose a significant financial burden on healthcare budgets and nations’ welfare [2].

Overweight, obesity, and low birth weight (LBW) are considered leading risk factors contributing to both the disease burden and the economic burden of NCDs. Currently, 69% of Saudi adults are overweight or obese, and obesity is the second risk factor for death in Saudis [3]. Saudi Arabia has one of the highest obesity rates globally (37.7% obesity prevalence, including 15.2% morbid obesity prevalence) [4]. The disease burden in Saudi Arabia and the population’s morbidity have increased due to high body mass index (BMI), high fasting plasma glucose concentration, high blood pressure, high LDL cholesterol, hypertension, and low physical activity [5]. Therefore, studying the methods used to identify overweight and obesity, such as anthropometric measures, may assist in determining and preventing NCDs and monitoring their trends.

The main anthropometric measurements that represent diagnostic criteria for obesity are height, weight, BMI, skinfold thickness, and body circumferences (waist, hip, and limb) [6]. Central (abdominal) obesity is an indicator of adverse health outcomes and is commonly estimated by measuring waist circumference. Ethnic-specific values for waist circumference have been established by the International Diabetes Federation (IDF), and it is widely used [7]. However, no specific cutoffs are available for Eastern Mediterranean and Middle East (Arab) populations due to the lack of sufficient data. The recommendation by the IDF is to use the European cutoffs: ≥94 cm (men) and ≥80 cm (women). The 2009 report on harmonizing the metabolic syndrome definitions still recommends using the IDF cutoffs for waist circumference in Middle Eastern populations [7].

Anthropometry is widely used in the field and clinical settings due to their simplicity, portability, inexpensiveness, and safety. Other anthropometrics are significant, such as facial anthropometrics, that have many practical applications for facial reconstructive procedures. In addition, stature, hand anthropometrics, and knee angle are essential in determining population cutoffs and are used in comparison studies. Anthropometric measurements are also commonly used in large population studies due to their convenience [8]. Thus, anthropometric measurements are significant in preventing health risks, determining health status, and improving individuals’ quality of life, which are all part of the Saudi 2030 Vision.

Saudi Vision 2030 was launched in 2016 [9] and is built on three fundamental pillars: A vibrant society, a thriving economy, and an ambitious nation. They function in harmony to achieve the desired objectives and maximize the benefits of the vision. The Health Sector Transformation Program has four objectives, including promoting the prevention of health risks [10]. This objective focuses on preventive health to address the key health challenges facing the population to reduce the health burdens of NCDs.

The concerning increase in the global prevalence of obesity and its related morbidity and mortality marks the significance of the assessment of body size, shape, and composition for health. Research efforts have progressed to understand obesity and despite these efforts, the global burden of obesity is increasing. A meta-analysis that included more than 300,000 people from multi-ethnic backgrounds, showed that the waist-to-height ratio provides a more robust tool for discriminating obesity-related cardiometabolic risk compared to BMI [11]. However, several studies showed different associations among age groups, gender, and ethnicity, highlighting the importance of age, gender, and ethnic-specific cutoff points [12,13,14,15]. There is possible evidence that Asians should have a lower waist circumference cutoff than Europeans [14,16]. The evidence is insufficient for specific cutoffs for African American, Hispanic, and Middle Eastern populations, but some studies indicate current cutoffs for Europeans may be appropriate [14]. This mandates the need for population-specific standardized cutoffs used in research methods to reduce discrepancies in findings that are partially due to different methods, to estimate the prevalence of overweight and obesity, predict health care costs, disease, and mortality. To ensure these measurements are taken precisely, clinicians must have standardized protocols and access to valid and reliable measures.

To our knowledge, there are no standard guidelines established for the Saudi population to evaluate anthropometric measurements. In Saudi adults, waist circumference is usually classified based on the classifications of International Diabetes Federation, but recent studies suggest the use of Saudi-specific cutoffs [17]. Overall, there is inconsistency in measuring and reporting of physical measurements among Saudi adults, which can be largely attributed to the lack of national guidelines.

Therefore, the aims of this review are to review the literature for studies conducted in Saudi Arabia that provide a national reference of anthropometric measurements for Saudi adults and elderly and to provide insights and recommendations to establish national guidelines and cutoffs in anthropometric measurements for Saudi adults/elderly that are in line with the Saudi vision 2030.

## 2. Materials and Methods

The search method has been reported previously [18]. A number of electronic databases were searched: Medline, PubMed, Saudi Digital Library, and Google Scholar from January 1990 to January 2021 [19]. Hand searches of reference lists of retrieved articles were undertaken. The search included the use of Medical Subject Headings (MeSH) and other keywords reported previously [18]. After screening titles and abstracts from the searched databases, 724 full articles were screened for eligibility. A total of 163 Saudi adult/elderly studies measured anthropometrics. All studies in the English language were included, as well as PhD theses and abstracts that provided sufficient information.

Exclusion criteria were reported previously [18]. Medical conditions, including diabetes, hypertension, and cardiovascular diseases, were included if they appeared during the search since they are the main comorbidities of obesity and are strongly related with BMI [20]. In addition, international anthropometric cutoffs are generated based on the prediction of these diseases [21]. Other studies on these diseases could be available but are not mentioned in this review since our search terms focused on anthropometrics [18] and not the medical conditions.

## 3. Measures and Anthropometrics of Saudis

### 3.1. Facial Anthropometrics

Several studies were conducted to assess Saudi facial anthropometry [22,23,24,25,26,27,28,29,30] that are essential for facial reconstructive procedures [25] and has many practical applications, including classification, diagnosis, and treatment of craniofacial anomalies, evaluation and treatment of post-traumatic deformities [29]. A study including 209 Saudi adults reported that nasal widths lie between Whites and Chinese. Compared to other ethnicities, the Saudi noses for both men and women have a larger nasofrontal angle. In contrast, the nasal tip angle/nasal tip protrusion in Saudi adults is significantly smaller than in other ethnicities [23]. These differences in facial anthropometrics with other ethnicities were consistent in some studies [25,26,30,31] that concluded differences in ethnic groups and that facial norms and measurements of other Caucasian populations cannot be applied to the Arab population. This distinction has a great impact on treatment planning for corrective, reconstructive, and orthognathic procedures. Furthermore, when facial anthropometrics were compared between five countries of the Gulf cooperation Council (Bahrain, Kuwait, Oman, Saudi Arabia, and the United Arab Emirates), Bahraini males had significantly wider mouths than Saudi Arabian males [28]. These results indicate that differences in facial anthropometrics exist between Arabs who are in the same ethnic group. Similarly, variations in body anthropometrics may be present between Arabs, suggesting specific cutoffs for each country [32].

### 3.2. Saudi Stature, Predictions from Hand and Foot Dimensions and Comparison to Other Populations

Stature is one of the most important elements in the identification of an individual [33]. Several Saudi adult studies estimated stature from hand dimensions [34,35,36,37]. A positive correlation was observed between hand length and palm length with stature [34,35,36]. In a more recent prospective study, 13 upper limb measurements were positively correlated with stature in Saudi men and the best single predictor was the bilateral ulnar length [37]. Three studies compared anthropometrics between the Saudi population and other populations [34,36,37]. The stature of Saudi Arabians was similar to the stature of Egyptians and Chinese, whereas Slovakians and Australian individuals were taller than Saudis. Indians and Bangladeshis individuals were shorter than Saudis [34]. When comparing hand measurements, Saudi Arabian measurements were similar to Chinese except for female hand length that was slightly longer in Saudis. Australians, Egyptians, and Nigerians had longer and wider hands in comparison with Saudis [34]. In contrast, another study found that the mean height of Indians was the longest in comparison to Saudis, Egyptians, and Filipinos. There was no significant difference between the Egyptian and Saudi Arabian groups regarding their stature and hand length [36]. Meta-analysis was conducted to compare Saudi stature and upper limb measurements with other populations. Cohen’s d method was used to quantify the effect sizes between Saudi men and the other populations. Relatively small differences in stature were observed between Saudi men and Egyptians or a mixed Turkish population, although significant differences were observed between Saudi men and northeastern Indians [37].

Recent studies assessed the correlation between height and foot length in Saudis [38]. The study concluded a direct relationship between foot length with the stature of Saudi adults [38]. Another study estimated stature from six lower limb dimensions in Saudi men [39]. All lower limb measurements showed statistically significant Pearson correlation coefficients with stature, and it was the strongest for tibial length. Comparisons of lower limb measurements between Saudis and other populations (Indians, Chinese, Italian, Australian, Turkish, Sudanese, and Egyptians) demonstrated a medium to large difference except for the high similarity in foot size with Slovak men [39]. The previous studies showed that stature is an important characteristic that reflects size, hence, these differences are an outcome of genetic/ethnic makeup which is also affected by physical activity, nutrition, and habits, such as shoe-wearing. Skeletal development is strongly governed by factors that differ between ethnicities, therefore, these models need to be population-specific.

### 3.3. Other Anthropometrics Assessed in Healthy Saudis

The availability of anthropometric data is significant to determine specific population cutoffs, to compare with other populations, and to link this data to population health. Thus, some researchers provided anthropometric data in Saudis regarding hand anthropometrics of engineering students that may aid in designing various devices and machines [40]. Another study aimed to establish foot posture index reference values in healthy Saudi adults that may be used in a range of clinical settings [41]. Furthermore, plastic and reconstructive surgeries in Saudi women have increased [42]. In addition, breast size may be a risk factor for developing breast cancer [43], which has been in an upward trend in Saudi Arabia [44], therefore, Al-Qattan and colleagues [45] identified the descriptive measurements of the breast in healthy Saudi women. Since the knee joint is the largest and one of the most complex joints of the body, Elfouhil et al. studied normal knee angles in Saudi adults and compared their values to other populations [46]. Two studies determined sex from hand dimensions and index/ring finger length ratio in Saudis [47] and radiographic measurements of the humerus [48].

Regarding comparative studies, a comparative study of palatal height and width between Saudi and Egyptian participants did not indicate an ethnic difference [49], whereas another comparative study found that Egyptians had longer limbs with relatively short torsos and rounded chest contours, and Saudis had shorter limbs with relatively longer torsos and flattened chest contours [50].

## 4. Results: Studies Conducted in Saudi Arabia Providing a National Reference of Anthropometric Measurements and Comparing Them with International Standards

Appendix A shows a total of (n = 151 studies) in adults/elderly and reports the reference of the cutoffs used in the studies and whether the studies mentioned how the anthropometrics were measured. Most studies used WHO cutoffs and the WHO measurement guideline [51,52,53]. Twelve studies in Table 1 are highlighted that contributed to the development of specific Saudi cutoffs.

Two studies on the Saudi population recommended higher waist circumference cutoffs in women but differed regarding the proposed cutoffs in men. The first study used data from the Saudi Abnormal Glucose Metabolism and Diabetes Impact Study (SAUDI-DM) to create waist circumference cutoffs used for metabolic syndrome definition [17]. The study was conducted from 2007 to 2009 and included 12,126 adults ≥18 years of age from 13 regions of Saudi Arabia. The study recommended to use lower cutoffs for men (92 cm) compared to the IDF, higher waist circumference in cutoff (87 cm) in women, and to use a BMI of 25 kg/m^2^ for men 28 kg/m^2^ for women when assessing the risk of metabolic syndrome [17]. There was no consideration for the ethnic diversity of the Saudi population.

The second study was a Ph.D. thesis that integrated data from five Saudi Surveys, three of which were conducted in the Riyadh region and included women (n = 13,117) median age (IQR): 39 (28–50) years, men (n = 10,851) median age (IQR): 40 (29–54) years [54]. Analyses of the data from five surveys suggested that waist circumference cutoffs in Saudis based on predicting comorbidity (hypertension, T2DM, dyslipidemia, any morbidity, any 2 comorbidities, and any 3 comorbidities) are as follows: 94–99 cm (men) and 90–92 cm (women). Additionally, analyses to explore BMI cutoffs suggested the women range from 28.2 to 29.7 (kg/m^2^) and 25.9 to 27.7 (kg/m^2^) for men. Future studies establishing waist circumference and waist-to-hip ratio cutoffs are recommended to follow the WHO guidelines [55] that are summarized in Appendix A.

Neck circumference has been a recent anthropometric in identifying overweight/obesity and cardiometabolic risk and cutoffs have been reported for Saudi adults [56,57]. A study on n = 700 Saudi adults (89% women) explored the association between neck circumference and multiple cardiometabolic risk factors. Data analyses were conducted on females only (n = 623) due to the small males’ sample size. In women, the study concluded that neck circumference ≥35.5 cm predicts ≥3 cardiometabolic risk factors. Moreover, the study explored cutoffs for BMI and waist circumferences and concluded that a BMI of 27.7 kg/m^2^ and waist circumference of 92 cm predicts ≥3 cardiometabolic risk factors in women. Results, however, cannot be generalized to the Riyadh population because participants were recruited from a hospital and primary care centers in Riyadh, 68% of the participants were 45 y or older, and 26% of the women were illiterate (compared to <10% at the national level between 2013 and 2017).

Since skinfold thickness is related with muscle mass [58], cutoffs for sarcopenia indices were developed for Saudi men [59]. However, the study was not representative of the Saudi population and included a small sample size. In addition, muscle mass between men and women is different [60] and studies including both sexes are essential to develop national cutoffs. In light of muscle mass, hand grip strength (HGS), and pinch strength are common measures to evaluate hand function and predict general health, therefore, two studies determined normative values of HGS and pinch strength in Saudi females [61] and elderly [62].

The established anthropometric cutoffs for the adult Saudi population from a representative sample include metabolic syndrome cutoffs [17,54]. Other established cutoffs from Saudi national surveys include BMI, waist circumference and waist-to-hip/height ratio [54]. However, these national cutoffs are rarely utilized, as shown in Appendix A. Therefore, it is essential to generalize and make these cutoffs easily accessible, such as providing them on the Saudi MOH website to be used by healthcare providers. Regarding anthropometric cutoffs for the elderly Saudi population, no cutoffs have been established from a representative sample (Table 1). 

**Table 1 healthcare-11-01982-t001:** Studies that contributed to the development of Saudi anthropometric cutoffs.

No	Author, Year (Reference)	Study Design	Region/City	Population Age	Sample n	Anthropometrics Assessed	Anthropometrics Assessment Definition	Comments	Established Cutoffs for Saudis
Adults									
1.	Almajwal et al., 2009 [63]	Cross-sectional	Eastern	≥30 years	197,681	Weight, height, BMI	WHO	Anthropometric measurement method mentioned. The study assessed the ability of BMI to diagnose obesity and to determine the optimal BMI cutoff points for the Saudi population based on the prevalence of diabetes and hypertension.	There is an increased risk of diabetes and hypertension relative to BMI, starting at a BMI as low as 21 kg/m^2^, but overall, there is no BMI cutoff with high predictive value for the development of these chronic diseases, including the WHO definition of obesity at BMI of 30 kg/m^2^.
2.	Albassam 2016 [64]	Cross-sectional	Riyadh	18–70 years	700	Weight, height, BMI, waist and neck circumference, % body fat	Based on references	Anthropometric measurement method mentioned.	The appropriate neck circumference to predict three or more metabolic risk factors in Saudi women is 35.5 cm.
3.	Al-Rubean et al., 2017 [17]	Cross-sectional	All regions	≥18 years	12,126	Weight, height, BMI, waist circumference	WHO/IDF	Anthropometric measurement method mentioned. The study identified the optimal cutoff values for anthropometrics for identifying the risk of metabolic syndrome.	The optimal cutoff values for identifying the risk of metabolic syndrome: ✓ WC, 92 cm for men and 87 cm for women ✓ WHR, 0.89 for men, 0.81 for women ✓ BMI, 25 kg/m^2^ for men, 28 kg/m^2^ for women
4.	Al-Kahtani 2017 [59]	Cross-sectional	Riyadh	20–35 years	232	Weight, height, waist circumference, body composition	Not mentioned	Anthropometric measurement method mentioned. The study identified the mean cutoff values for sarcopenia indices in Saudi men.	✓ The reference value of appendicular lean mass /height squared (ALM/ht2), was 8.97 kg/m^2^ ✓ Hand grip strength measured 42.8 ± 7.6 kg The cutoff values for sarcopenia indices for Saudi young men are different from those of other ethnicities
5.	Alfadhli et al., 2017 [56]	Cross-sectional	Madinah	≥18 years	785	Weight, height, waist and neck circumference	NCEP ATP III	Anthropometric measurement method mentioned. The study determined the optimal cutoff value for neck circumference to identify. overweight/obesity and predict cardiometabolic risk.	Neck circumference cutoffs for identifying participants with central obesity ✓ ≥39.25 cm for men ✓ ≥34.75 cm for women
6.	Alkhalaf 2017, [54]	Cross-sectional	Saudi national surveys	Adults	23,968	Weight, height, BMI, waist circumference, waist-to-hip ratio waist to height ratio, body composition	WHO and Several references	Ph.D. thesis (chapters 5–7). Anthropometric measurement method mentioned. Author provided several tables of diagnostic performance of anthropometrics in predicting health morbidities in Arab adults. The study suggested new cutoff values for BMI, waist circumference, waist-to-hip/height ratio for Saudi adults.	WC cutoffs for Saudis ✓ 90 to 92 cm (women) ✓ 94 to 99 cm (men) %SMM cutoffs for Saudis ✓ 29 to 32% for men ✓ 26 to 28% for women BMI cutoff points for Saudis ✓ 28.2 to 29.7 (kg/m^2^) for women ✓ 25.9 to 27.7 (kg/m^2^) for men WHtR cutoff points for Saudis ✓ 0.57 to 0.59 for women ✓ 0.56 to 0.59 for men WHR cutoff points for Saudis ✓ 0.82 to 0.85 for women ✓ 0.98 to 1.0 for men
7.	Alzeidan et al., 2019 [57]	Cross-sectional	Riyadh	18–85 years	3063	Weight, height, BMI, neck, hip, and waist circumference	Based on several references	Anthropometric measurement method mentioned. The study showed that neck circumference was a predictor of obesity and metabolic syndrome. The study provided neck circumference cutoffs that predict obesity.	NC cutoffs to predict obesity ✓ ≥37.5 cm for men ✓ ≥32.5 cm for women
8.	Al-Hanawi et al., 2020 [65]	Cross-sectional	All regions	>15 years	7746	Weight, height, BMI	Natural log of BMI	Anthropometric measurement method referred to a reference. Representative sample from the Saudi Health Interview survey. The study decomposed the BMI gender gap into its associated factors across the entire BMI distribution by using counterfactual regression. methods. Females showed a higher BMI than males.	The study used new distribution-based regression methods to explain the BMI gender gap. The advantage of this method is that the study observed heterogeneity in how determinants are associated with BMI differentials at various points of distribution.
9.	Almousa 2021 [66]	Cross-sectional	4 regions	18–63	1074	Weight, height, BMI, and other anthropometrics	ISO/ASTM standard	3D body scanner	In this study, the first anthropometric database for the Saudi female population was established using 3D body scanning technology, and a sizing system for this target population was developed.
10.	Shaheen et al., 2021 [61]	Cross-sectional	Riyadh	19–25 years	139	Weight, height, BMI	Not mentioned	Anthropometric measurement method mentioned.	This study established the hand grip strength and pinch strengths normative values and developed the prediction equations in a sample of healthy female college students.
**Elderly**									
11.	Alqahtani et al., 2019 [62]	Cross-sectional	Riyadh	65–80 years	1048	Weight, height, BMI, arm circumference	Not mentioned	Anthropometric measurement method mentioned.	This study is the first that established normative values of hand grip for older adults in Saudi Arabia.
12.	Bindawas et al., 2019 [67]	Cross-sectional	Riyadh	≥60 years	2045	Weight, height, BMI, handgrip strength	Compared hand grip strength to other populations		The study established normative data for handgrip strength in older Saudi adults.

BMI: Body mass index, ISO: International Organization for Standardization, NC: Neck circumference, NCEP-ATP III: National Cholesterol Education Program’s Adult Treatment Panel, %SMM: Percentage of skeletal muscle mass to body weight, WC: Waist circumference, WHR: Waist-to-hip ratio, WHtR: Waist-to-height ratio, WHO: World Health Organization.

## 5. Discussion

We identified nationally representative new cutoffs of BMI, waist circumference, and metabolic syndrome for Saudi adults (Table 1). Although neck circumference has not been fully established for Saudi adults, several studies showed positive potential for developing specific cutoffs to predict overweight/obesity and metabolic syndrome. Anthropometric cutoffs for Saudi elderly are not well established, and more studies are needed.

### 5.1. Recommendations for Establishing Saudi Guidelines of Anthropometric Measurements and Directions for Future Work in Line with the Saudi 2030 Vision

Specific cutoffs have been created for Saudi adults >19 years from representative samples for BMI, waist circumference, and metabolic syndrome [17,54] (discussed in Section 4) (Table 1). However, generalization and endorsement by the Saudi MOH are necessary for these cutoffs to be used and to make practitioners aware of their availability. For adults aged >19 years, two methods can be applied to ensure that anthropometrics are measured on a regular basis. First, all adults not attending college or university should be registered in their local health clinics or general practice clinic. Once they are registered, annual reminders may be sent to individuals by Absher platform, an electronic Saudi platform for several e-services offered by the Ministry of Interior and its sectors for Saudis and non-Saudis living in Saudi Arabia, to visit the clinic for screening and for physical measurements. A standardized protocol on the measurement method should be available, Saudi adult cutoffs should be used, and documentation of these anthropometrics is necessary. The annual documentation of anthropometric measurement may help in showing trends of weight change that may help physicians in predicting NCDs risks, such as diabetes and hypertension. These trends may be prevented by intervening with lifestyle changes, referring to dietitians/physicians, or enrolling in weight loss programs.

The second method to ensure that anthropometrics of adults attending college/university are measured annually is by the collaboration between the Saudi MOH and the Saudi Ministry of Education. A comparative study was conducted to compare Arab countries in the quantity and quality of obesity research [68]. The Arab countries contribute to 1.0% of the global research output on obesity, and Saudi Arabia ranked the 39th country. This shows the need for more research in Saudi Arabia. Saudi vision 2030 aims to build an educational system aligned with market needs and enhance research activities and innovations [9]. A brief report discussed potential methods to enhance the research activities in Saudi Arabia to improve the Saudi rank to be in the top 10 countries in the Global Competitiveness Index, which is one of the objectives of the Saudi 2030 vision [69]. In line with this, this review suggests that annual screening of anthropometric measurements in education sectors (colleges/universities) that help in detecting obesity and their comorbidities is essential. Training students of medical fields to use standardized methods of anthropometric measurements and Saudi cutoffs will enable them to strengthen their clinical skills. This is in line with the Saudi Vision 2030 objectives that aim to build an educational system aligned with market needs.

Furthermore, education sectors, specifically universities, should initiate research centers and create journals for their students. Several journals are available in Saudi universities, but many universities still lack the availability of their own journals [69]. The anthropometric data can be published in several online sources for health statistics in Saudi Arabia [19], which will increase Saudi Arabia’s research output and help in the rank of the Global Competitiveness Index.

Studies on the elderly are limited (Table 1 and Appendix A), and more Saudi representative studies are needed. First, the age of the elderly needs to be clearer for the Saudi population. The WHO estimates that in 2019, the number of people aged 60 years and older was 1 billion. This number will increase to 1.4 billion by 2030 and 2.1 billion by 2050 [70]. Healthy aging is defined by the WHO as “developing and maintaining the functional ability that enables well-being in older age”. The WHO provides a guideline on integrated care for older people [71] that may help in managing malnutrition and, consequently, may increase life expectancy.

After defining the age of elderly in the Saudi population, annual screening is critical to prevent malnutrition and muscle loss and to study the trends in the Saudi elderly. Collecting a sample of representative Saudi elderly is essential in determining their health status and determinants. Specifically, sarcopenia is important to assess in elderly people because it is an age-related muscular disease manifesting as a loss of muscle function that may lead to detrimental consequences. Few studies have been conducted to measure sarcopenia in the Saudi elderly [59], however, a recent protocol aims to measure the prevalence of sarcopenia in the Saudi elderly and its association with lifestyle behaviors [72]. Annual screening of malnutrition and muscle mass is essential to determine and prevent sarcopenia. A standard guideline for the elderly can be made to increase muscle mass from exercise experts and physical therapists with dietary recommendations as to how to increase protein intake from registered dietitians. This guideline can be made readable and in Arabic with many visuals for the elderly and their caregivers (Table 2).

### 5.2. General Recommendations and Directions for Future Work

The cost of medical care is dramatically increasing in Saudi Arabia, and the budget of the Ministry of Health has increased by 300% in the last 13 years and is further increasing due to COVID-19 negative effects [73]. One of the reasons for the increased cost of medical care may be due to the unavailability of a national health record system used by all health institutions leading to the repetition of medical examinations, lab, and diagnostic tests. A proposed method is to link medical health records to the national ID number for Saudis or to the resident number for non-Saudis. After linking medical health records to ID numbers, patients will have a complete history of all medical examinations, lab results, and surgeries. By endorsing the utilization of the national health record system by all healthcare institutions in Saudi Arabia, patients may conduct their medical examinations in any institution without repetition, consequently reducing medical costs. Furthermore, healthcare providers in any healthcare institution will have a complete medical history of the patients, which will assist them in formulating trends of disease and anthropometrics throughout patients’ lives.

Diet is considered a modifiable factor that affects anthropometric measurements and body composition based on a meta-analysis [74]. However, few studies have been conducted to study this relationship in Saudis. The first step in studying the relationship between diet and anthropometric measurements is to collect information on the diet of Saudis from nationally representative samples. Few studies have assessed the diet of Saudis from nationally representative samples [75], and more studies are needed. It is essential to create a national diet and nutrition survey (NDNS) similar to the one in the UK [76] that is accessible to conduct statistical analyses and explore the relationship between different food items and dietary habits with anthropometric measurements and body composition. The NDNS may collect information on physical activity since it is considered another modifiable factor that impacts anthropometric outcomes [77], and only 17.40% of Saudis perform the recommended amount of physical activity [78]. These data will be beneficial for researchers and are in line with Saudi 2030 Vision which aims to increase public participation in sports and athletic activities to promote a healthy lifestyle [9].

Several anthropometric traits are heritable according to twin studies that showed 40–80% contribution of genetic factors for BMI [79] and up to 80% for height [80]. A more recent meta-analysis identified six novel loci for body shape using a combination of anthropometric traits (body mass index, height, weight, waist, and hip circumference, waist-to-hip ratio). The results showed that body shape is heritable and associated with cardiometabolic outcomes [81]. Another meta-analysis showed associations between copy number variants on anthropometric traits and that anthropometric traits share genetic loci with developmental and psychiatric disorders [82]. These studies show that studying the associations between genes and anthropometric traits is significant and may predict cardiometabolic outcomes and some psychiatric disorders.

One of the Saudi Vision 2030 projects in healthcare is the Saudi Genome Program [9]. The Saudi Genome program has five objectives, including building a Saudi genetic database, enabling scientists and researchers to benefit from a database of genetic information, studying genetic variants, and developing diagnostic and preventive tools to reduce the incidence of genetic diseases. This program may be used to identify genes, loci, and copy number variants associated with anthropometric traits in a representative Saudi population. These studies may help in predicting cardiometabolic outcomes and other diseases. Consequently, developing diagnostic and preventive tools to reduce the incidence of these diseases in genetically susceptible individuals may help in reducing the burden of NCDs in Saudi Arabia.

**Table 2 healthcare-11-01982-t002:** Summary of recommendations for establishing Saudi guidelines of anthropometric measurements of adults/elderly and directions for future work in line with the Saudi 2030 Vision.

Recommendations- Adults/Elderly	Directions for Future Work
1. Cutoffs of BMI, waist circumference and metabolic syndrome for Saudi adults >19 years from a representative sample [17,54] are available (see Section 4) (Table 1)	Generalize, publish the Saudi adult cutoffs, and endorse them to be used by the Saudi MOH. Adults not attending college/university may register in their local health clinic. Health clinics may send annual reminders to individuals through Absher or mobile number for general health screening and anthropometric measurement. Use the WHO guideline on how to measure adult anthropometrics to ensure standardization [83]. May use 3D human shapes, which are non-invasive and more accessible for 17 anthropometric measurements [84]. Develop a protocol on anthropometric data documentation in health clinics and usage of the adult Saudi cutoffs and their interpretation. Anthropometrics of adults attending college/university can be measured annually at their college/university by the collaboration between Saudi MOH and Saudi Ministry of Education. Developing and establishing research guidelines in colleges/universities for anthropometric studies [85]. Anthropometric data can be published in university journals or other Saudi journals [19,69] to increase research output and the Global Competitiveness Index in line with Saudi 2030 Vision.
2. Create standard cutoffs of BMI, waist circumference, skinfold thickness, muscle and fat mass and hand grip strength for Saudi elderly	Define the age of elderly in the Saudi population and generalize it. Collect a representative sample of Saudi elderly.
3. Create a standard guideline on how to prevent malnutrition, loss of muscle mass, and increase protein intake of Saudi elderly in Arabic	Collaboration between physical therapists, dietitians, and physicians to create the guideline, publish it, and endorse it by the Saudi MOH. Annual screening of muscle mass, sarcopenia [72], and malnutrition for the elderly to study the trends of Saudi elderly. Use the WHO guideline to manage malnutrition in the elderly [71].
**Recommendations for individuals with health conditions and disabilities**	**Directions for future work**
4. Create screening programs for health conditions that are related to anthropometrics similar to Singapore [86], obesity, hypertension, diabetes mellitus, and hyperlipidemia	Establish a database of the population’s anthropometrics and explore their relation to obesity, hypertension, diabetes mellitus, and hyperlipidemia to facilitate identifying trends over the years, causes, and implications for policy makers and preventive measures needed
**General recommendations and directions for future work**	
5. Link medical health records from all medical facilities in Saudi Arabia with the Saudi national ID or resident ID number for non-Saudis to establish national data. 6. After linking medical health records with ID number, create a unified national health record system used by all health institutions in Saudi Arabia [10,87] and report to the MOH similar to the NHS used in the UK [88]. This national health record system will include anthropometrics from birth, diagnoses, lab tests, medications, and surgeries. The national health record system may decrease the repetition of medical examinations and lab tests. In addition, it assists healthcare providers in formulating trends of disease and anthropometrics for patients. The data from the national health record system will help in identifying the prevalence, trends, the causes of diseases and conducting preventive measures. 7. Diet is a modifiable factor that affects anthropometric measurements and body composition [74]. Collecting the diet of Saudis from nationally representative samples [75] and creating a national diet and nutrition survey (NDNS) similar to the one in the UK [76] is essential. The NDNS may collect physical activity data [78] that is accessible to conduct statistical analyses and explore the relationship between different food items, dietary habits and physical activity with anthropometric measurements, body composition, and NCDs. 8. Study the associations between genes and anthropometric traits and their association with cardiometabolic outcomes and other diseases in a representative sample of the Saudi population as conducted in two meta-analyses [81,82]. These associations are in line with the Saudi Genome Program in the 2030 Saudi vision [9].	

MOH; Ministry of Health, NCDs: Non-communicable diseases, NDNS: National Diet and Nutrition Survey, WHO: World Health Organization.

## 6. Conclusions

NCDs cause financial burden [2] and are attributed to 71% of global deaths [1]. Studying the tools used to detect NCDs is necessary. Anthropometric measurements are the first step in determining health status and assist in detecting NCDs. In conclusion, this review has summarized the evidence of anthropometric measurements in Saudi adults/elderly. Cutoffs for metabolic syndrome, waist circumference, and BMI were developed for Saudi adults, however, they are not used nor endorsed to be used by the MOH. More studies are needed to assess the reasons for not using Saudi adults’ cutoffs. Limited studies were found in the Saudi elderly; therefore, more studies are needed. This review provides recommendations for establishing Saudi guidelines of anthropometric measurements and directions for future work in line with the Saudi 2030 Vision. The information presented in this review will aid in establishing reports and a standardized protocol for anthropometric measurements and their acceptable cutoffs that can assist in the early detection and prevention of NCDs in Saudi Arabia.

## Data Availability

No new data was created or analyzed in this study. Data sharing is not applicable to this article.

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
