# Peer review of "A Review of Anthropometric Measurements for Saudi Adults and Elderly, Directions for Future Work and Recommendations to Establish Saudi Guidelines in Line with the Saudi 2030 Vision"

_healthcare, 2023, doi:10.3390/healthcare11141982_

Round 1

Reviewer 1 Report

The paper titled “Setting Standard Limits for Anthropometric Measurements Help Detect Noncommunicable Diseases: A Review 3 of Anthropometric Measurements for Saudi Adults and Elderly” written by Essra A. Noorwali and Abeer M. Aljaadi is well written and easy to read fluent. There is coherence between the objectives and the methodology. The results are clearly expressed and adequately discussed. In my opinion, the work is publishable incorporating some changes.

The authors reviewed Saudi studies and guidelines to provide information and recommendations for establishing national guidelines on anthropometric measurements for Saudi adults and elderly individuals.

While the introduction highlights the importance of knowing the cutoffs used for anthropometric measurements in the diagnosis of health conditions such as overweight, obesity, diabetes, cardiovascular diseases, high blood pressure, and high LDL cholesterol, the materials and methods section presents results related to anthopometry of face, upper limb, and height, among others (see sections 3.1, 3.2, and 3.3). In my opinion, the authors should cover this aspect in the introduction; otherwise, when it appears later in the results, it may be difficult for the reader to understand its relevance.

One important point that I would like to highlight is the use of "race" to refer to the difference between Saudis and other countries. Since the 1950s and 1960s, the term "race" has been questioned with the advent of human genetics and new anthropological currents. Today, there is broad scientific consensus that there are no human races in a biological sense, and the use of that terminology is outdated. Although its use (especially outside the academic field) is still common, the correct term to use is ethnicity. Roth and Ivermak define race as a cognitive structure that divides people into inherent categories based on phenotypic characteristics, while Cornell and Hartmann, and Roth define ethnicity as a cognitive structure that divides people based on common ancestry, shared history, and cultural heritage. The meaning of ethnicity is broader and takes into account various aspects of people, not only phenotypic characteristics.

Unfortunately I couldn't check every reference but I did a general search and there are highlighted  those who that caught my attention

References

Roth, W. D. & Ivemark, B. Genetic options: The impact of genetic ancestry testing o consumers’ racial and ethnic identities. Am. J. Sociol. 124(1), 150–184. https:// doi. org/ 10. 1086/ 697487 (2018).

Cornell, S. & Hartmann, D. Ethnicity and Race: Making Identities in a Changing World (Pine Forge Press, 1998).

Roth, W. Race Migrations: Latinos and the Cultural Transformation of Race (Stanford University Press, 2012).

Author Response

The paper titled “Setting Standard Limits for Anthropometric Measurements Help Detect Noncommunicable Diseases: A Review of Anthropometric Measurements for Saudi Adults and Elderly” written by Essra A. Noorwali and Abeer M. Aljaadi is well written and easy to read fluent. There is coherence between the objectives and the methodology. The results are clearly expressed and adequately discussed. In my opinion, the work is publishable incorporating some changes.

Thank you for your comment and for taking the time to review our manuscript. We will address the changes as shown below.

The authors reviewed Saudi studies and guidelines to provide information and recommendations for establishing national guidelines on anthropometric measurements for Saudi adults and elderly individuals.

While the introduction highlights the importance of knowing the cutoffs used for anthropometric measurements in the diagnosis of health conditions such as overweight, obesity, diabetes, cardiovascular diseases, high blood pressure, and high LDL cholesterol, the materials and methods section presents results related to anthopometry of face, upper limb, and height, among others (see sections 3.1, 3.2, and 3.3). In my opinion, the authors should cover this aspect in the introduction; otherwise, when it appears later in the results, it may be difficult for the reader to understand its relevance.

This has now been covered in the introduction, see lines 55-58.

One important point that I would like to highlight is the use of "race" to refer to the difference between Saudis and other countries. Since the 1950s and 1960s, the term "race" has been questioned with the advent of human genetics and new anthropological currents. Today, there is broad scientific consensus that there are no human races in a biological sense, and the use of that terminology is outdated. Although its use (especially outside the academic field) is still common, the correct term to use is ethnicity. Roth and Ivermak define race as a cognitive structure that divides people into inherent categories based on phenotypic characteristics, while Cornell and Hartmann, and Roth define ethnicity as a cognitive structure that divides people based on common ancestry, shared history, and cultural heritage. The meaning of ethnicity is broader and takes into account various aspects of people, not only phenotypic characteristics.

Unfortunately I couldn't check every reference but I did a general search and there are highlighted  those who that caught my attention

 Thank you for this valuable comment. We have now changed the word race to ethnicity throughout the whole manuscript (see lines 122-125).

References

Roth, W. D. & Ivemark, B. Genetic options: The impact of genetic ancestry testing o consumers’ racial and ethnic identities. Am. J. Sociol. 124(1), 150–184. https:// doi. org/ 10. 1086/ 697487 (2018).

Cornell, S. & Hartmann, D. Ethnicity and Race: Making Identities in a Changing World (Pine Forge Press, 1998).

Roth, W. Race Migrations: Latinos and the Cultural Transformation of Race (Stanford University Press, 2012).

Comments on the Quality of English Language

We appreciate this feedback; edits have been incorporated.

Submission Date

12 April 2023

Date of this review

21 Apr 2023 21:54:19

Reviewer 2 Report

The proposal to review Saudi studies on the anthropometry of its population is interesting and valuable.

The title of the study is “Establishing standard cutoffs for anthropometric measurements assist in detecting noncommunicable diseases: A review of anthropometric measurements for Saudi adults and elderly”… however the conclusions are different.

The proposal to review Saudi studies on the anthropometry of its population is interesting and valuable.

The authors highlight the need for studies that define reference values ​​for the Saudi population, which in my opinion is extremely valuable for national health policies.

However, the format in which the data are presented in Table 1 is not very didactic. 163 Saudi studies were listed and looking at the form presented, little is used objectively.

My conclusion is that the studies were mostly conducted within the World Health Organization reference values ​​as they should be.

Therefore, the listed works have the value in front of what was proposed in their objectives.

An important message from the authors must be considered: the importance of anthropometric reference parameters for the Saudi population.

The authors list the following study information: Author, Year (reference), Region/City, Population age, Sample n, Anthropometrics studies, Anthropometrics assessment definition and Comments individually.

Another important point. As the study appears to be a systematic review, it would be important for the authors to use an appropriate methodology applied to a systematic review.

Minor editing of English language required

Author Response

The proposal to review Saudi studies on the anthropometry of its population is interesting and valuable.

Thank you for your interest and for taking the time to review our manuscript.

The title of the study is “Establishing standard cutoffs for anthropometric measurements assist in detecting noncommunicable diseases: A review of anthropometric measurements for Saudi adults and elderly”… however the conclusions are different.

Thank you for your comment. In response, we have changed the title to: A review of anthropometric measurements for Saudi adults and elderly, directions for future work and recommendations to establish Saudi guidelines in line with the Saudi 2030 Vision.

The proposal to review Saudi studies on the anthropometry of its population is interesting and valuable.

The authors highlight the need for studies that define reference values ​​for the Saudi population, which in my opinion is extremely valuable for national health policies.

However, the format in which the data are presented in Table 1 is not very didactic. 163 Saudi studies were listed and looking at the form presented, little is used objectively.

Thank you for your comment. In response we have moved table 1 to supplementary material (see Table S1) and added table 1 in the result section that shows only the established cut-offs for Saudis. In addition, table S2 was added based on the WHO guidelines for developing population specific cut-off points that can be used by Saudis when developing new cut-offs.

My conclusion is that the studies were mostly conducted within the World Health Organization reference values ​​as they should be.

Therefore, the listed works have the value in front of what was proposed in their objectives.

Most studies have used the WHO reference values however, the WHO Expert consultation on Obesity (2000a) stated the “need to develop sex-specific waist circumference cut-off points appropriate for different populations”. The report provided a table of sex-specific waist circumference and risk of metabolic complications associated with obesity in Caucasians from a random sample in the Netherlands. However, this table as reported by WHO is just an example and not the WHO recommendations. Therefore, specific population cut-off points are recommended based on the WHO guidelines (Table S2).

An important message from the authors must be considered: the importance of anthropometric reference parameters for the Saudi population.

We agree that we should consider the importance of anthropometric reference parameters for the Saudi population. Therefore, we added the following paragraph (see lines 42-52, lines 72-80).

“Central (abdominal) obesity is an indicator of adverse health outcomes and is commonly estimated by measuring waist circumference. Ethnic specific values for waist circumference have been established by the International Diabetes Federation (IDF) and it is widely used [53]. However, no specific cutoffs are available for Eastern Mediterranean and Middle East (Arab) populations due to the lack of sufficient data. The recommendation by the IDF is to use the European cutoffs: ≥ 94 cm (men) and ≥ 80 cm (women). The 2009 report on harmonizing the metabolic syndrome definitions still recommends using the IDF cutoffs for waist circumference in Middle Eastern populations [53].   ”

“A meta-analysis which included more than 300,000 people from multi-ethnic backgrounds showed that waist-to height-ratio provides a more robust tool for discriminating obesity-related cardiometabolic risk compared to BMI [10]. However, several studies showed different associations among age groups, gender and ethnicity highlighting the importance of age, gender and ethnic-specific cut-off points [11][12][13][14]. There is possible evidence that Asians should have a lower waist circumference cutoff than Europeans [13,15]. The evidence is insufficient for specific cutoffs for African American, Hispanic and Middle Eastern populations, but some studies indicate current cutoffs for Europeans may be appropriate [13]. “

The authors list the following study information: Author, Year (reference), Region/City, Population age, Sample n, Anthropometrics studies, Anthropometrics assessment definition and Comments individually.

Another important point. As the study appears to be a systematic review, it would be important for the authors to use an appropriate methodology applied to a systematic review.

Thank you for your comment. It is a systematic search and not a systematic review since we did not apply quality assessment of the included studies. Our justification is that we needed to include all the studies that conducted anthropometric measurements to have an overview of what is available and to be able to provide recommendations and directions for future work.

Comments on the Quality of English Language

Minor editing of English language required

We have fixed some typos.

Submission Date

12 April 2023

Date of this review

26 Apr 2023 14:43:59

Reviewer 3 Report

The article concerns the establishing of standard cut-off points for the anthropometric examination of adult and elderly Saudi populations. Authors may consider the following comments:

One of the main objections is the layout of the article. This, however, results with indirect and misleading conception.

Firstly, the authors present some assumptions regarding the relationship between anthropometry and noncommunicable diseases, then they presented a 15-page long table. Ultimately, they refer to various, for example, dietary or genetic relationships with anthropometric differences. Finally, they mentioned the cut-off points presented in Table 1 of Section 4 - which in fact do not exist. After going through the manuscript, it is difficult to determine what the final conclusion actually is.

Methodological error - Gender cannot be ignored in an anthropometric study - studies that do not take into account this natural law should be removed from the presented work.

Summary: In my opinion, the work should be rejected in its current form because it does not represent any significant scientific value. The table and footnotes should be shortened only to the most necessary data. Then, from these data, perform meta-analysis, test power assessment, ROC curves to examine the optimal cut-off points of individual elements of the anthropometric study for the prevalence of individual cardiovascular diseases - e.g. abdominal circumference and hypertension or metabolic syndrome. With that, such a work would be what the title actually stands for.

Additionally, the work is difficult to comprehend. Sentences are long and incomprehensible, requiring careful reanalysis.

I.e. Section of introduction - In Line 69-70

-The whole paragraph should be rephrased and reformatted. Numbers indicating next points should be removed and reformatted in such a way to highlight important points in different sentences. Continuous sentences without any full stop punctuation or period are very insensible.

Section of method and materials, line 82- 83.

-Sentence should be rephrased. The word “even” can be changed to inclusive to or as well as.

Section 3. Measures and anthropometrics of Saudis – line 136-137

-Sentences can be amended as it is expressed in an unnecessary way.

Author Response

The article concerns the establishing of standard cut-off points for the anthropometric examination of adult and elderly Saudi populations. Authors may consider the following comments:

One of the main objections is the layout of the article. This, however, results with indirect and misleading conception.

Firstly, the authors present some assumptions regarding the relationship between anthropometry and noncommunicable diseases, then they presented a 15-page long table.

Thank you for your comment. The relationship between anthropometry and noncommunicable diseases is not an assumption. Anthropometric cutoff points are developed based on their risk with diseases as reported by the WHO (World Health Organization). In response, we have added table S2 that outlines guidelines and recommendations by the WHO for developing anthropometric cutoff points.

Ultimately, they refer to various, for example, dietary or genetic relationships with anthropometric differences. Finally, they mentioned the cut-off points presented in Table 1 of Section 4 - which in fact do not exist. After going through the manuscript, it is difficult to determine what the final conclusion actually is.

We appreciate your feedback. In response, we have created a new table with the most important studies that contributed to the development of specific Saudi cutoff points (Table 1). Some adjustments have been made for more clarity. The conclusion is in line with the objectives of the review: cutoffs for Saudi adults for BMI, metabolic syndrome and waist circumference have been created; however, need endorsement by the Ministry of Health to be used. In addition, the second objective of this review is to provide guidelines and recommendations that will help in the establishment of a national protocol regarding anthropometric measurements. This review provided the resources (references) that can be used and provided practical methods that can be applied (See table S2 and Table 2).

Methodological error - Gender cannot be ignored in an anthropometric study - studies that do not take into account this natural law should be removed from the presented work.

Studies included have considered sex/gender. Specifically, the studies that contributed to the development of Saudi anthropometric cut-offs have provided cutoffs for both men and women (shown in Table 1).

Summary: In my opinion, the work should be rejected in its current form because it does not represent any significant scientific value. The table and footnotes should be shortened only to the most necessary data. We now have moved the long table to the supplementary material (see table S1) and only included the studies that contributed to the development of Saudi anthropometric cutoffs (see table 1). We hope this makes it more concise and helps with readability.

Then, from these data, perform meta-analysis, test power assessment, ROC curves to examine the optimal cut-off points of individual elements of the anthropometric study for the prevalence of individual cardiovascular diseases - e.g. abdominal circumference and hypertension or metabolic syndrome. With that, such a work would be what the title actually stands for.

Thank you for this suggestion. The objectives of this review were to review the literature of Saudi national guidelines and studies previously conducted in Saudi Arabia and to provide insights and recommendations to establish national guidelines in anthropometric measurements for Saudi adults/elderly in line with the Saudi 2030 Vision. In response, we have added table 1 that outlines the developed Saudi anthropometric cutoffs, the studies in table 1 have used ROC curves to identify the optimal cutoffs. In addition, table S2 have been added that provides recommendations from the WHO regarding developing anthropometric cutoff points. Table S2 highlights some limitations of the ROC curves that were also outlined by the WHO (see https://www.who.int/publications/i/item/9789241501491 ).

This review was necessary to be conducted since it is the first review to identify Saudi adult/elderly anthropometrics based on a systematic search. A meta-analysis with an aim to examine the optimal cutoff points would be ideal after this review. Our work was the first step in identifying the available anthropometrics for Saudis and is extremely valuable for national health policies. We agree that the title might have been misleading, therefore in response, we have modified the title.

Comments on the Quality of English Language

Additionally, the work is difficult to comprehend. Sentences are long and incomprehensible, requiring careful reanalysis.

I.e. Section of introduction - In Line 69-70

-The whole paragraph should be rephrased and reformatted. Numbers indicating next points should be removed and reformatted in such a way to highlight important points in different sentences. Continuous sentences without any full stop punctuation or period are very insensible.

This has been edited.

Section of method and materials, line 82- 83.

-Sentence should be rephrased. The word “even” can be changed to inclusive to or as well as.

The word “even” has been changed (see lines 107-108).

Section 3. Measures and anthropometrics of Saudis – line 136-137

-Sentences can be amended as it is expressed in an unnecessary way.

This has been edited (see line 165).

Submission Date

12 April 2023

Date of this review

08 Jun 2023 17:54:37

Reviewer 4 Report

This study aims to review the literature of Saudi national guidelines and studies previously conducted in Saudi Arabia  and to provide insights and recommendations to establish national guidelines in anthropometric measurements for Saudi adults/elderly. The argument is relevant in the field, but does not clearly fill in the gaps. Compared to the previous published material, the proposal adds to establish guidelines.

The title creates expectations that are not met. The authors have done a long job of enumeration of all the works published in a span of 31 years, concerning anthropometric measurements, but then they do not present usable results. Table 1 is the long list of articles studied. Table 2 foresees the proposals for the future. A table with any cutoffs is missing. Authors should include a table with cutoffs, as related in section 4 What further controls should be considered? 

They should insert a table reporting what is described in section 4 (Results), considering that in Discussion the Authors state “We identified nationally representative new cutoffs of BMI, waist circumference and metabolic syndrome for Saudi adults”.

In Conclusion the Authors say “This review provides recommendations for establishing Saudi guidelines of anthropometric measurements ..” therefore all the paper is summarized in Table 2 ? 

The conclusions should be improved because they don't answer the main question.

References: many references are incomplete, have no volume and pages, some are unobtainable and therefore unverifiable. In some cases, names or titles are capitalized. 

Many references are not traceable on the common Pubmed system. Proceedings are often cited and should be replaced by an article published in a journal. The rules for Authors do not seem to be respected.

Author Response

This study aims to review the literature of Saudi national guidelines and studies previously conducted in Saudi Arabia and to provide insights and recommendations to establish national guidelines in anthropometric measurements for Saudi adults/elderly. The argument is relevant in the field but does not clearly fill in the gaps. Compared to the previous published material, the proposal adds to establish guidelines.

Thank you for your interest and for taking the time to review our manuscript. Yes, one of the aims of the review is to establish guidelines (see table S2 and table 2), which have not been reported before.

The title creates expectations that are not met.

We agree that the title might have been misleading therefore in response, we have changed the title.

The authors have done a long job of enumeration of all the works published in a span of 31 years, concerning anthropometric measurements, but then they do not present usable results. Table 1 is the long list of articles studied. Table 2 foresees the proposals for the future. A table with any cutoffs is missing. Authors should include a table with cutoffs, as related in section 4 What further controls should be considered? 

Thank you for your comment. In response to this comment, we have moved table 1 to supplementary material (see Table S1) and added table 1 in the result section that shows only the established cut-offs for Saudis. In addition, table S2 was added based on the WHO guidelines in developing population specific cut-off points that can be used by Saudis when developing new cut-offs.

They should insert a table reporting what is described in section 4 (Results), considering that in Discussion the Authors state “We identified nationally representative new cutoffs of BMI, waist circumference and metabolic syndrome for Saudi adults”.

We agree. In response, table 1 has been added.

In Conclusion the Authors say “This review provides recommendations for establishing Saudi guidelines of anthropometric measurements ..” therefore all the paper is summarized in Table 2 ? The conclusions should be improved because they don't answer the main question.

Amendments have been made to the conclusion. The conclusion is in line with the two main objectives of the review.

First objective: (1) to review the literature for studies conducted in Saudi Arabia that provide a national reference of anthropometric measurements for Saudi adults and elderly.

The result of this objective: is table S1, table 1 and the result section.

The conclusion for this objective was mentioned “this review has summarized the evidence of anthropometric measurements in Saudi adults/elderly. Cutoffs for metabolic syndrome, waist circumference and BMI were developed for Saudi adults, however, they are not used nor endorsed    by the MOH. More studies are needed to assess the reasons for not using Saudi adults’ cutoffs. Limited studies were found in Saudi elderly and more studies are needed”.

Second objective (2) to provide insights and recommendations regarding the resources that can be used to establish national guidelines and cutoffs of anthropometric measurements for Saudi adults/elderly that are in line with the Saudi vision 2030.

The result of this objective: table S2 and table 2.

The conclusion for this objective: “This review provides recommendations for establishing Saudi guidelines of anthropometric measurements and directions for future work in line with the Saudi 2030 Vision. This review will help policy makers and the Saudi MOH to establish reports and a standardized protocol to be used in Saudi Arabia for anthropometric measurements and their valid cutoffs that may assist in detecting NCDs and preventing them”.

References: many references are incomplete, have no volume and pages, some are unobtainable and therefore unverifiable. In some cases, names or titles are capitalized. 

Thank you for your comment. This will be amended in the final version.

Many references are not traceable on the common Pubmed system. Proceedings are often cited and should be replaced by an article published in a journal. The rules for Authors do not seem to be respected.

Thank you for your comment. When an article is available, the proceeding is replaced. Some references were obtained from the Saudi Digital Library and might not show in PubMed. References will be edited in the final version.

Submission Date

12 April 2023

Date of this review

06 Jun 2023 10:03:16

Round 2

Reviewer 3 Report

Lne 90  add space.

Summary:

In its current form, the work is much more legible, simpler and clearer. Especially the additional column in the abbreviated table corresponds to the assumptions of the work and gives the basic and most important data. After minor corrections in spaces and scoring, the work can be considered for publication.

As above

Author Response

Lne 90  add space

Thank you for reviewing the manuscript. Space has been added in line 90. 

In its current form, the work is much more legible, simpler and clearer. Especially the additional column in the abbreviated table corresponds to the assumptions of the work and gives the basic and most important data. After minor corrections in spaces and scoring, the work can be considered for publication.

We appreciate your valuable comments that have made the manuscript clearer and provided the most important data.